# Antifungal Activity of *Guiera senegalensis*: From the Chemical Composition to the Mitochondrial Toxic Effects and Tyrosinase Inhibition

**DOI:** 10.3390/antibiotics12050869

**Published:** 2023-05-08

**Authors:** Rute Moreira, Federico Ferreres, Ángel Gil-Izquierdo, Nelson G. M. Gomes, Luísa Araújo, Eugénia Pinto, Paula B. Andrade, Romeu A. Videira

**Affiliations:** 1REQUIMTE/LAQV, Laboratório de Farmacognosia, Departamento de Química, Faculdade de Farmácia, Universidade do Porto, 4050-313 Porto, Portugal; up201408307@edu.ff.up.pt (R.M.); ngomes@ff.up.pt (N.G.M.G.); pandrade@ff.up.pt (P.B.A.); 2Molecular Recognition and Encapsulation (REM) Group, Department of Food Technology and Nutrition, Universidad Católica de Murcia, 30107 Murcia, Spain; fferreres@ucam.edu; 3Research Group on Quality, Safety and Bioactivity of Plant Foods, Department of Food Science and Technology, CEBAS (CSIC), Campus University Espinardo, 30100 Murcia, Spain; angelgil@cebas.csic.es; 4MDS—Medicamentos e Diagnósticos em Saúde, Avenida dos Combatentes da Liberdade da Pátria, Bissau, Guinea-Bissau; luisaaraujo@quilaban.pt; 5Laboratory of Microbiology, Biological Sciences Department, Faculty of Pharmacy, University of Porto, 4050-313 Porto, Portugal; epinto@ff.up.pt; 6Interdisciplinary Centre of Marine and Environmental Research (CIIMAR), University of Porto, 4450-208 Matosinhos, Portugal

**Keywords:** *Guiera senegalensis*, antifungal activity, mitochondria, tyrosinase, yeasts

## Abstract

Pest resistance against fungicides is a widespread and increasing problem, with impact on crop production and public health, making the development of new fungicides an urgent need. Chemical analyses of a crude methanol extract (CME) of *Guiera senegalensis* leaves revealed the presence of sugars, phospholipids, phytosterols, guieranone A, porphyrin-containing compounds, and phenolics. To connect chemical composition with biological effects, solid-phase extraction was used to discard water-soluble compounds with low affinity for the C18 matrix and obtain an ethyl acetate fraction (EAF) that concentrates guieranone A and chlorophylls, and a methanol fraction (MF) dominated by phenolics. While the CME and MF exhibited poor antifungal activity against *Aspergillus fumigatus*, *Fusarium oxysporum* and *Colletotrichum gloeosporioides*, the EAF demonstrated antifungal activity against these filamentous fungi, particularly against *C. gloeosporioides*. Studies with yeasts revealed that the EAF has strong effectiveness against *Saccharomyces cerevisiae*, *Cryptococcus neoformans* and *Candida krusei* with MICs of 8, 8 and 16 μg/mL, respectively. A combination of in vivo and in vitro studies shows that the EAF can function as a mitochondrial toxin, compromising complexes I and II activities, and as a strong inhibitor of fungal tyrosinase (Ki = 14.40 ± 4.49 µg/mL). Thus, EAF appears to be a promising candidate for the development of new multi-target fungicides.

## 1. Introduction

Fungicides are the most used pesticides around the world, representing about 40% of all pesticide application in the European Union [1]. In addition to their use in plant pest control, fungicides are also used to treat human and animal fungal infections, which are nowadays considered an important health problem, due to the increasing number of immunocompromised patients [2]. Additionally, the continued use of antimycotics and the application of fungicides in agriculture closely related to therapeutic drugs brings an inevitable issue: the emergence of drug-resistant strains [3]. The acquired resistance, as well as the emergence of fungal strains with higher pathogenicity and the upgraded ability to spread around the world makes the search for new antifungal agents an urgent need. Therefore, the research programs developed to characterize the bioactivity of medicinal plant extracts have been broadened to include the assessment of antifungal activity [2].

*Guiera senegalensis* J. F. Gmel. is a medicinal shrub widely distributed in African countries, particularly in semi-desert areas of the Sudano–Sahelian region. In African traditional medicine, this plant is commonly used for the treatment of several chronic diseases, colds, bronchitis, stomach pain, vomiting, jaundice, kidney stones and infections. A decoction of its leaves has been used in the treatment of venereal diseases as an antimicrobial agent [4]. Bosisio and colleagues [5] reported the antifungal properties of aqueous and methanol leaves extracts, assessing their effects on six species of yeast and twelve species of filamentous fungi. Other parts of *G. senegalensis* were studied concerning their antifungal potential, with the finding that the extracts from galls and roots have activity against five serotype strains of fungi and three strains of yeast, respectively [6,7]. While the relationship between the antifungal activity and the chemical profile of the above-mentioned plant extracts was not established, another study supports the idea that the antifungal activity of *G. senegalensis* extracts can be attributed to guieranone A [4]. However, the mechanisms of action underlying guieranone A antifungal activity remain to be clarified. Furthermore, the presence of toxic alkaloids has been identified in different parts of *G. senegalensis*, including in the leaves and roots [8]. Thus, the valorization of *G. senegalensis* for new fungicide development requires the establishment of clear relationships between the chemical composition of the plant extracts and the antifungal activity.

Working within this scientific scenario, a crude methanol extract (CME) of *G. senegalensis* leaves was prepared and fractionated on solid phase C18 column to obtain an ethyl acetate fraction (EAF) and a methanol fraction (MF). The chemical composition of CME, EAF and MF will be disclosed and their relationship with antifungal activity against several filamentous fungi and yeast strains will be discussed in the context of pest control and human health. Therefore, the present work aims to characterize if and how *G. senegalensis* leaves have antifungal activity by revealing the mechanisms of action and establishing a clear relationship between activity and chemical composition. This is the key to establishing a scientific basis for the use of *G. senegalensis* leaves as a renewable source for the development of new fungicides.

## 2. Results and Discussion

### 2.1. Chemical Composition of G. senegalensis CME, MF and EAF

The maceration of leaves of *G. senegalensis* in methanol allowed the preparation of CME with an extraction yield of 17.58 ± 5.43%. A solid-phase extraction procedure (SPE) on a Chromabond C18 non-end-capped column was used for CME fractionation, allowing to discard the water-soluble compounds and to collect the lipophilic components with ethyl acetate, the EAF, and the remaining compounds with methanol, the MF. The obtained fraction yields for the EAF and MF were 24.15 ± 16.43% and 11.44 ± 6.89%, respectively (Table 1). The general chemical characterization of the CME and of its fractions followed a work plan to assess the total levels of Folin–Ciocalteu reactive phenolic compounds, sugars, phospholipids, phytosterols, chlorophylls and alkaloids, with subsequent determination of the enrichment factor for these parameters in each fraction (Table 1).

Reduction of the Folin–Ciocalteu reagent (FCR) in the presence of phenolic compounds results in the production of molybdenum-tungsten blue that is measured spectrophotometrically at 760 nm [9]. Using this methodology, the concentration of Folin–Ciocalteu reactive phenolic compounds in *G. senegalensis* CME was 37.63 ± 3.48 µg/mg extract, expressed as gallic acid equivalents. The compounds with this type of antioxidant activity were mainly retained in the EAF, since this fraction exhibited a concentration of 46.12 ± 5.98 µg/mg extract and an enrichment factor of 1.23, whereas the MF showed a concentration of only 17.7 ± 1.13 µg/mg extract with an enrichment factor of 0.47. These data indicate that many phenolic compounds present in CME exhibit lipophilic properties. Previously, it was shown that the poly-methoxylated flavonoids are dragged from the Chromabond C18 column by low polar solvents such as ethyl acetate [10], suggesting that the EAF may contain poly-methoxylated compounds.

Total sugars content was determined spectrophotometrically at 490 nm after a reaction with phenol in the presence of sulfuric acid [11]. The levels of total sugars were higher in the EAF (76.45 ± 20.90 µg glucose equivalents per mg extract) than in the CME (58.91 ± 4.56 µg glucose equivalents per mg extract), with an enrichment factor of 1.30. On the other hand, the MF, with 33.70 ± 3.44 µg glucose equivalents per mg extract, had a lower content than the CME (enrichment factor of 0.57). It is important to highlight that a significant portion of the sugars (close to 50%) present in *G. senegalensis* CME were discarded by water used in the clean-up procedure step of the SPE. Thus, the sugars quantified in the two fractions may include free sugars that were not removed by the water, as well as the sugars naturally attached to the phenolic compounds.

Phospholipids were quantitated spectrophotometrically at 800 nm by phosphorous assay, after acidic hydrolysis [12]. The total phospholipid content in the CME, EAF and MF was 37.70 ± 6.02, 22.38 ± 6.20 and 8.45 ± 6.38 µg PC (C16:0/18:1) equivalents per mg extract, respectively. Thus, the enrichment factor in both fractions is less than unity, indicating that most phospholipids are retained by the silica in the column.

Phytosterols were quantified by the Liebermann–Burchard assay, taking advantage of the blue-green colour product generated by the reaction of sterols with acetic anhydride in the presence of concentrated sulfuric acid [13]. The results in Table 1 show that the *G. senegalensis* EAF had a phytosterol content of 193.65 ± 20.38 µg stigmasterol equivalents per mg extract and an enrichment factor of 2.87, which represents 69% of the total phytosterols present in the CME (67.39 ± 6.30 µg stigmasterol equivalents per mg extract). With 17.17 ± 8.16 µg stigmasterol equivalents per mg extract, the MF only retained 2.8% of the sterols present in the CME. These data indicate that a significant part of the phytosterols remained in the SPE column.

The chlorophylls content of the *G. senegalensis* CME, EAF and MF was assessed by the chlorophylls’ UV-visible spectrum signature [14]. The chlorophylls content in the CME was 5.83 ± 1.0 µg chlorophyll *a* equivalents per mg extract, whereas in the MF, they were not detected. In fact, all chlorophylls were dragged with ethyl acetate. Therefore, their content in the EAF was 28.63 ± 1.36 µg chlorophyll *a* equivalents/mg extract, corresponding to an enrichment factor of 4.91.

Several studies report the presence of β-carboline alkaloids in different parts of *G. senegalensis*, including in the leaves [8,15,16]. Thus, a preliminary screening for the presence of this class of compounds was carried out with the alkaloids reagents (Dragendorff, Mayer and Bertrand) in the *G. senegalensis* CME, EAF and MF, and the results were negative. Previous studies that report the presence of alkaloids in *G. senegalensis* are based on extracts that are generally obtained with solvents with lower polarity, such as dichloromethane and chloroform [8,15]. In fact, HPLC analysis of an alkaloid-directed extract obtained from leaves or from the solid residue of the CME confirmed the presence of these compounds in the *G. senegalensis* leaves and their absence in the CME (Appendix A).

### 2.2. Characterization of the Phenolic Profile of the G. senegalensis CME

The HPLC-DAD-ESI/MS^n^ analysis of the CME obtained from the leaves of *G. senegalensis* evidences a rich phenolic profile, as shown in UV chromatograms recorded at 280 and 340 nm (Figure 1) with the peak identifications shown in Table 2. Combining UV spectrum features of each chromatogram peak with its MS data, it was possible to identify ten galloyl derivatives (**1**–**10**), three flavonol glycosides-cinnamoyl derivatives (**29**–**31**), seventeen flavonoid glycosides and three aglycones (**11**–**28**, **32** and **33**). Additionally, guieranone A was identified as peak **34**, while two chromatogram peaks (**35**, **36**) were not identified (Table 2 and Figure 1).

Regarding galloyl derivatives, compounds **1**–**10** present a similar UV spectrum (~272–276 nm) and specific MS signatures that allow their identification, as will be described below. Compounds **1** and **4** (Rt 2.2 and 3.1 min), both with [M−H]^−^ at *m*/*z* 343.0673, are isomers (C_14_H_15_O_10_) that exhibit an MS fragmentation pattern with ions at *m*/*z* 191.0565 and 169.0145, corresponding to the deprotonated molecular ions of quinic and gallic acids, respectively. As such, it is suggested that they should be galloylquinic acid isomers. Compounds **2**, **5**, **6** and **8** (Rt 2.4, 5.0, 6.6 and 6.9 min) with ([M−H]^−^) at *m*/*z* 495.0780 are isomers (C_21_H_19_O_14_), and their MS fragmentations revealed ions at *m*/*z* 343.0669, 191.0568 and 169.0149, corresponding to deprotonated molecular ions of galloylquinic, quinic and gallic acids, respectively.

The deprotonated molecular ion of **3** (Rt 2.9 min) at *m*/*z* 169.0147 (C_7_H_6_O_5_) exhibited an MS fragmentation with a carbonyl loss (−44 amu, CO_2_) detected by the ion at *m*/*z* 125.0244 ([C_6_H_5_O_3_]^−^) and, therefore, corresponds to gallic acid. Compounds **7** and **9** (Rt 6.8 and 9.1 min) have deprotonated molecular ions with an *m*/*z* of 647.0892, which is 152 amu higher than compounds **2**, **5**, **6** and **8**. They exhibited an MS fragmentation pattern with *m*/*z* species resulting from the losses of one and two gallic acid units (152), with or without water accompaniment, in addition to the deprotonated gallic acid ion; therefore, they can be identified as tri-galloylquinic acid isomers. Compound **10** (Rt 10.1 min; [M−H]^−^ at *m*/*z* 799.1000) is another additional gallic acid since its MS fragmentation profile exhibited ions resulting from the losses of different fragments of gallic acid and the base peak of the ions was at *m*/*z* 601.0818 [(tGQA-H)-(W+28)]^−^, so the compound can be identified as tetra-galloylquinic acid. Many of the ions of the MS^2^ fragmentations of **7**, **9** and **10** resulting from losses of one or more galloyl radicals accompanied by water are not indicated in Table 2 due to the large number and variability of their presence.

Regarding flavonol glycosides and aglycones, compounds **11**–**28** and **33** exhibit UV spectra characteristic of flavones and **32** of flavanone (Table 2). Compounds **28** and **33** have a UV band I (~370 nm) typical of flavonols with a free 3-hydroxyl, and their masses correspond to quercetin (3,5,7,3′,4′-pentahydroxyflavone) and isorhamnetin (3,5,7,4′-tetrahydroxy-3′-methoxyflavone), respectively. Additionally, the UV spectrum of **25** could not be properly resolved due to the co-elution with **26**, but its mass ([M−H]^−^ at *m*/*z* 317.0307) corresponds to myricetin (3,5,7,3′,4′,5′-hexahydroxyflavone). The remaining constituents that display UV spectra characteristic of flavonols with a substituted 3-hydroxyl (Table 2) are characterized below. In the MS fragmentation profile of **11**, **12**, **16** and **17** there is the loss of a 152 amu fragment (radical galloyl) to give base peaks, whose masses in **11** and **12** correspond to myricetin-3-*O*-hexoside, i.e., the [M−H]^−^ of **13** and **14**, and in **16** and **17** to quercetin-3-*O*-hexoside, i.e., the [M−H]^−^ of **20** and **21** (Table 2). Thus, these compounds can be identified as myricetin-3-*O*-(galloyl)galactoside (**11**), myricetin-3-*O*-(galloyl)glucoside (**12**), quercetin-3-*O*-(galloyl)galactoside (**16**), and quercetin-3-*O*-(galloyl)glucoside (**17**).

Compounds **13**–**15**, **18**–**24**, **26** and **27** exhibit deprotonated molecular ions of monoglycosides, with base peaks of [myricetin-2H]^−^ (**13–15**, **18** and **19**), [quercetin-H]^−^ (**20**–**24**) and [kaempferol-H/2H]^−^ (**26** and **27**) observed in their MS fragmentations (Table 2). Therefore, **13** and **14** correspond to myricetin-3-*O*-hexoside isomers and, considering their elution order in RP, they can be labelled as myricetin-3-*O*-galactoside and myricetin-3-*O*-glucoside, respectively; **15** and **18** are myricetin-3-*O*-pentoside isomers: myricetin-3-*O*-xyloside (**15**) and myricetin-3-*O*-arabinoside (**18**). Compound **19** corresponds to myricetin-3-*O*-rhamnoside; quercetin-3-*O*-hexoside isomers (**20** and **21**) are quercetin-3-*O*-galactoside and quercetin-3-*O*-glucoside; quercetin-3-*O*-pentoside isomers (**22** and **23**) are quercetin-3-*O*-xyloside and quercetin-3-*O*-arabinoside; **24** is quercetin-3-*O*-rhamnoside; **26** and **27** are kaempferol-3-*O*-pentoside and kaempferol-3-*O*-rhamnoside.

Compounds **29**–**31** (Rt of 18.3, 18.5 and 18.7 min, respectively) are isomers with the characteristic UV spectrum of flavonoids acylated with cinnamoyl acids since UV overlap derived from the flavonoid moiety and the cinnamoyl acid (UV: 266, 296sh, 314, 350sh nm) is observed (Table 2). In the MS fragmentations of their deprotonated molecular ions at *m*/*z* 593.1301 ([C_30_H_25_O_13_]^−^), loss of a *p*-coumaroyl radical (−146 amu) is observed to give place to an ion at *m*/*z* 447.0935 ([C_21_H_19_O_11_]^−^) and the deprotonated ion of the aglycone at *m*/*z* 284.0338 ([M-2H, C_15_H_8_O_6_]^−^), which indicates kaempferol. As such, **29**–**31** can be labelled as kaempferol-3-*O*-(*p*-coumaroyl)hexoside isomers.

The UV spectrum of **32** (Rt 21.5 min) suggests the presence of a flavanone backbone (UV: 285, 332sh nm). Its deprotonated molecular ion at *m*/*z* 301.0728 ([C_16_H_14_O_6_]^−^) matches that of hesperetin (5,7,3′-trihydroxy-4′-methoxy-flavanone) (Table 2).

Compounds **34**–**36** are very lipophilic, eluting at the end of the chromatogram (Rt 23.1, 23.7 and 26.1 min), and display similar UV spectra (266sh, 274, 312, 326, 380 nm; 276, 306, 360 nm and 268, 278sh, 312, 326 nm). Compound **34** ionizes in positive mode [M+H]^+^ ion at *m*/*z* 317.1290 (C_18_H_20_O_5_) and exhibits an MS fragmentation pattern compatible with that previously reported for guieranone A [4] since its characteristics acylium ion fragments are detected at *m*/*z* 287 [C_17_H_19_O_4_]^+^, 275 [C_15_H_15_O_5_]^+^ and at *m*/*z* 69 [C4H_5_O]^+^, as shown in the insert of Figure 2.

As will be described below, this compound was also detected in the EAF, and its MS^2^ spectrum is displayed in the insert of Figure 2. Compounds **35** and **36** also ionize in positive mode with [M+H]^+^ ions at *m*/*z* 301.1069 (C_17_H_16_O_5_) and at *m*/*z* 335.1486 (C_18_H_22_O_6_), respectively. However, their weak MS fragmentation does not allow structure elucidation.

Regarding the phenolic profile of *G. senegalensis* leaves, the identification of galloyl derivatives (**1**–**10**) (Table 2) in the CME is not surprising as the occurrence of gallic acid and a series of galloylquinic acid isomers was previously reported [17,18,19,20]. Myricetin, quercetin and kaempferol derivatives are also known to occur in the species, as reported for samples of leaves and galls of plants obtained in Burkina Faso, Senegal, and Sudan [18,19,21,22,23]. However, the present work reports a detailed phenolic characterization of samples collected in Guinea-Bissau that confirms the previously known phenolic profile of the species. The phenolic profile is enlarged with the identification, for the first time, of two myricetin derivatives (**11**, **15**), two quercetin derivatives (**16**, **17**) and, at least, two kaempferol derivatives (**29**–**31**).

The compound quantification was also performed by HPLC-DAD using the reference compounds and standard curves indicated in Appendix A. The concentration of each identified compound in the *G. senegalensis* CME is displayed in the last column of Table 2. The quantitative analysis revealed that flavonol-3-*O*-glycosides were predominant in the *G. senegalensis* CME, comprising ca. 85% of the total quantifiable phenolic content. Among the flavonol-3-*O*-glycosides, myricetin-3-*O*-rhamnoside (**19**: 36.94 ± 1.21 g/Kg dry extract) and myricetin-3-*O*-glucoside (**14**: 29.4 ± 1.39 g/Kg dry extract) were the major compounds, accounting for ca. 27% of the total phenolic content. It is worth highlighting the significant content of galloyl derivatives (**1**–**10**: 33.76 ± 0.50 g/Kg dry extract). They constituted ca. 9% of the total quantifiable phenolic content and included gallic acid (**3**) and the tetra-galloyl derivative (**10**). These were the most abundant compounds of the phenolic acid class. Additionally, the flavonols glycosides-cinnamoyl derivatives (**29**–**31**) represented about 4% of the total phenolic content, with kaempferol-3-*O*-(*p*-coumaroyl)hexoside isomer (**30**) being the most representative. Guieranone A, a naphthyl butanone, was, after myricetin-3-*O*-rhamnoside, the major compound with 36.46 ± 2.87 g/Kg dry extract and represented ca. 14% of the total content (Table 2).

### 2.3. Characterization of the Phenolic and Chlorophyll Profiles of the G. senegalensis EAF

Following the above-mentioned strategy, the *G. senegalensis* EAF was analysed to identify and quantify the phenolic compounds retained in this fraction. The HPLC-DAD analysis with a UV chromatogram recorded at 280 nm is displayed in Figure 2. The figure also shows the concentrations determined for the identified compounds and the MS/MS fragmentation pattern obtained for peak **11**, which was identified as guieranone A. Eleven compounds, all identified in the *G. senegalensis* CME, were detected in the EAF: gallic acid (**1**), five flavonoid glycosides (**2**–**5**, **7**), one aglycone (**6**), three flavonol glycosides-cinnamoyl derivatives (**8**–**10**), and guieranone A (**11**) (a naphthyl butanone). Concerning the quantification, guieranone A (**11**) was the major compound (293.35 ± 11.35 g/Kg dry extract) in the *G. senegalensis* EAF, representing ca. 76% of the total compounds identified under these conditions. Myricetin-3-*O*-glucoside (**2**: 40.51 ± 2.31 g/Kg dry extract), the flavonol glycoside-cinnamoyl derivative (**10**: 21.61 ± 0.51 g/Kg dry extract), and myricetin-3-*O*-xyloside (**3**: 16.32 ± 11.35 g/Kg dry extract) also exhibited significant relative abundance. The other seven compounds identified appeared in minor concentrations.

The *G. senegalensis* EAF was also analysed by means of HPLC-DAD-ESI/MS^n^ to promote the separation and identification of compounds with higher hydrophilicity such as chlorophylls (C30 YMC column, using *tert*-butyl methyl ether and methanol as mobile phases). The HPLC chromatogram recorded at 420 nm is displayed in Figure 3. Combining the UV spectrum features with MS data enabled the identification of chlorophyll *a* (**4**) and six compounds that are derivatives of chlorophyll *a* (**1**, **3**, **5**, **7**, **10**, **11**), as well as chlorophyll *b* (**2**) and three additional structurally related compounds (**6**, **8**, **9**), as indicated in Table 3. Thus, the compounds were quantified by HPLC-DAD using chlorophyll *a* and *b* as reference compounds and the standard curves indicated in Appendix A. As will be described below, the diversity of chlorophyll-related compounds detected in the EAF may result from the expected chlorophyll catabolism during leaf senescence triggered by the drying process.

Chlorophylls and their derivatives, analysed in positive ion mode, exhibit a well-defined fragmentation pattern with product ions resulting from unimolecular heterolytic decomposition of peripheral substituents [24]. Thus, MS signatures used to identify the compounds are the product ions that emerge from: (i) the replacement of central magnesium by two protons, (ii) the loss of the phytyl chain followed by the total or partial fragmentation of the propionic chain, and (iii) the fragmentation of the β-keto ester group.

Accordingly, compounds **2** (Rt 11.9 min) and **4** (Rt 18.8 min) with [M+H]^+^ ions at *m*/*z* 907.55 and 893.70 were identified as chlorophylls *b* and *a*, respectively, since they exhibited an MS fragmentation with a loss of the phytyl group (−278 amu), followed by a propionic acid (−74 amu), which were detected by the ions at *m*/*z* 553.1992 (C_32_H_25_MgN_4_O_4_) and 539.2418 (C_32_H_27_MgN_4_O_3_), respectively (Table 3). For both, the detected ion at *m*/*z* 227.1603 can be assigned to an additional phytyl fragmentation. Derivatives of the chlorophylls *b* and *a* resulting from the replacement of central magnesium by two protons (pheophytins) were detected as compounds **9** (Rt 34.88 min) and **11** (Rt 36.95 min) with [M+H]^+^ ions at 885.55 and 871.57, respectively. Pheophytins exhibit a characteristic fragmentation pattern of chlorophylls. For example, pheophytin *a*, detected by an ion at *m*/*z* 871.57, produces MS^2^ fragmentations with ions at *m*/*z* 593.2754 (chlorophyll *a* − Mg − phytyl), 533.2546 (chlorophyll *a* − Mg − phytil − CH_3_COOH), and at 519.2418 (chlorophyll a − Mg − phytyl − CH_3_CH_2_COOH). In pheophytin *b*, an additional fragmentation pathway that starts with the loss of a β-keto ester group and follows with fragmentation of the phytyl chain is detected, as suggested by ions at *m*/*z* 826.5379 (chlorophyll *b* − Mg − CH_3_COO) and 547.2350 (chlorophyll *b* − Mg − CH_2_COOH − phytyl). As previously reported, the replacement of the central magnesium atom of the tetrapyrrol ring of chlorophylls by reactions involving a metal-chelating substance is a very common catabolic process during leaf senescence [25]. Thus, the concentrations of the pheophytins *a* and *b* in the EAF were significantly higher than in the native compounds (chlorophyll *a* 1.07 ± 0.18, pheophytin *a* 44.76 ± 1.78 g/Kg dry extract; chlorophyll *b* 0.67 ± 0.03, pheophytin *b* 12.67 ± 1.82 g/kg dry extract).

Native chlorophylls and pheophytins are sensitive to oxidation processes that mainly generate two types of products. The main modification involves the replacement of the H atom at C13^2^ by a hydroxyl group, producing the hydroxyl derivatives [24]. In fact, hydroxyl derivatives of chlorophyll *a* and pheophytin *a* are detected as compounds **3** (Rt 13.44 min) and **10** (Rt 35.38 min), respectively, with [M+H]^+^ at *m*/*z* 909.53 and 887.65. Compound **8** (Rt 32.60 min), with an *m*/*z* at 901.65, was identified as HO-pheophytin *b*. As indicated in Table 3, the concentrations of these oxidized compounds decrease in the series HO-chlorophyll *a* > HO-pheophytin *a* > HO-pheophytin *b*. The other type of oxidation reactions occurs at C15^1^ with the formation of a lactone group and formation of hydroxy-lactone derivatives. Compounds **1** (Rt 7.71 min, *m*/*z* at 925.00), **6** (Rt 25.68 min, *m*/*z* at 917.70) and **7** (Rt 27.27 min, *m*/*z* at 903.65) were identified as HO-lactone-chlorophyll *a*, HO-lactone-pheophytin *b* and HO-lactone-pheophytin *a*, respectively. The concentration of OH-lactone-pheophytin *a* was higher than the other HO-lactone derivatives, but significantly lower than hydroxyl derivatives (Table 3). Compound **5**, Rt 21.19 min and *m*/*z* at 593.27, exhibited MS^2^ fragmentation ions compatible with the structure of pheophorbide *a*, as suggested by the ion with *m*/*z* at 533.2544 (pheophytin *a* − phytyl − CH_2_COOH). The concentration of this dephytylated catabolite of chlorophyll *a* in the EAF was 1.52 ± 0.40 g/Kg of dry extract.

### 2.4. Fungal Susceptibility to G. senegalensis CME, MF and EAF

The antifungal activity of *G. senegalensis* CME and its fractions (MF and EAF) on filamentous fungi was investigated by using species that affect agricultural crops and/or cause human and animal mycoses. The effects of the CME, MF and EAF on *Fusarium oxysporum*, *Aspergillus fumigatus*, *Colletotrichum gloeosporioides*, assessed by the minimal inhibitory concentration (MIC) and minimal fungicidal concentration (MFC), are displayed in Table 4. Although the CME and MF did not exhibit antifungal activity against these filamentous fungi at concentrations below 2000 µg/mL, the EAF showed antifungal activity against all species. Moreover, the EAF was particularly active against *C. gloeosporioides* since a MIC value of 500 and a MFC value of 1000 µg/mL were determined (Table 4). These results indicate the biological relevance of the *G. senegalensis* EAF since *C. gloeosporioides* is a prevalent pathogen of many economically important crops, including cashew, mango, papaya, avocado, and citrus [26,27], and this pathogen has developed resistance against many of the available fungicides [28].

Considering the positive results of the EAF towards filamentous fungi, the panel was broadened to include the human pathogenic yeasts (*Cryptococcus neoformans*, *Candida albicans*, and *Candida krusei*) and *Sacharomyces cerevisiae*, a non-pathogenic species used to allow the investigation of its mode of action with safety. The MIC and MFC values of the *G. senegalensis* EAF on the above-mentioned yeasts are displayed in Table 5, as well as the results obtained with penconazole and mancozeb, used as reference fungicides. The data in Table 5 show that the EAF exhibited strong antifungal activity against the pathogenic strains *C. neoformans* (MIC = 8 µg/mL, MFC = 125 µg/mL) and *C. krusei* (MIC = 16 µg/mL, MFC = 500 µg/mL), as well as against *S. cerevisiae* (MIC = 8 µg/mL, MFC = 500 µg/mL). Under the same assay conditions, the antifungal activities of penconazole and fluconazole against *C. neoformans* were characterized by MIC values of 16 and 8 µg/mL and MFC values of >16 and >32 µg/mL, respectively. Therefore, the antifungal power of the *G. senegalensis* EAF against *C. neoformans* is in the same order of magnitude as penconazole, a fungicide available for use in pest management, and as fluconazole, a fungicide used in clinical settings.

The antifungal activity of *G. senegalensis* CME and its fractions was also evaluated using cultures of *S. cerevisiae* and *C. neoformans* in liquid medium supplemented with 50 µg/mL of *G. senegalensis* leaves CME (filled squares), MF (open triangles) or EAF (open circles) (Figure 4A,B).

As a control, yeast cultures were grown in media without extracts, and the increase in the optical density at 610 nm was used to follow the time-dependent growth of yeast strains at 30 °C during 60 h. The long time intervals without recordings of optical density correspond to the night cycles. According to the results of Figure 4A,B, the growth of *S. cerevisiae* and *C. neoformans* in the control condition can be described by an exponential function, with a specific growth rate of 0.0447 and 0.0315 h^−1^, respectively. While MF did not induce detectable effects, exposure to the CME or EAF at 50 µg/mL led to strong inhibitory effects on the growth of both yeast strains. For example, the CME promoted a decrease in the specific growth rate of *S. cerevisiae* to 33% of the control and of *C. neoformans* to 40% of the control. On the other hand, in the presence of the same concentration of EAF, the specific growth rate of *S. cerevisiae* decreased to 30% of the control and that of *C. neoformans* decreased to 33% of the control. Therefore, the effects on the growth curves of yeast strains are in line with the results obtained in the MIC and MFC susceptibility tests (Table 5). These results confirm that the compounds with lipophilic properties underlie the antifungal activity of the CMEobtained from *G. senegalensis*; therefore, they are retained in the EAF during the fractionation procedure. Figure 4C,D show the effects of increasing concentrations of the EAF on the growth of *S. cerevisiae* and *C. neoformans* cultures, with the values of specific growth rates indicated in the inserts. As indicated by the decrease in the specific growth rate (*S. cerevisiae*: EC_50_ = 37.44 µg/mL; *C. neoformans:* EC_50_ = 39.00 µg/mL) as well as by the low values of optical density at the end of the assay, the inhibitory effects of the EAF towards both yeast strains are concentration-dependent.

In a previous study, it was reported that the ethanol extract of *G. senegalensis* leaves exhibited antifungal activity against *Cladosporium cucumerinum*. This antifungal activity was ascribed to guieranone A, as revealed by assays with the isolated compound [4]. Thus, the antifungal activity of the *G. senegalensis* leaves CME detected in the present work, which was conserved and/or amplified in the EAF (Table 4 and Table 5; Figure 4), suggests that the antifungal activity against the filamentous fungi and yeast strains can be also attributed to guieranone A. Moreover, the fraction with the weakest antifungal activity was *G. senegalensis* MF, in which guieranone A was not detected. This result excludes phenolic compounds from a role in this activity. However, the contribution of other compounds to the antifungal activity revealed by *G. senegalensis* leaves CME cannot be ruled out since the EAF fraction is rich in chlorophyll-related compounds, which have well-known antifungal properties [29,30]. Therefore, the contribution of guieranone A and chlorophyll-related compounds to the antifungal activity of the EAF is an open question with scientific and pharmacological relevance. In fact, additive and synergistic effects, as well as potentiation and antagonism can play key roles in the activity of a mixture of compounds, including plant extracts [31]. Thus, the minor components of the EAF may also contribute to the detected antifungal activity.

### 2.5. G. senegalensis EAF Exhibits Antifungal Activity Related to Mitochondrial Dysfunction

Many of the commercially profitable fungicides exhibit mechanisms of action that compromise the functionality of fungi mitochondria. In general, these compounds work as inhibitors of the mitochondrial redox chain complexes or as lipophilic compounds that affect the functional organization of the inner membrane, promoting disruption of the mitochondrial membrane potential [32]. Thus, the mechanisms underlying the antifungal activity of the *G. senegalensis* EAF were investigated, considering the effects on the functionality of the mitochondrial redox chain complexes, using *S. cerevisiae* as model organism. Yeast cultures of *S. cerevisiae* treated with 50 µg/mL of *G. senegalensis* EAF for 8 h and non-treated controls were used to isolate mitochondria and assess the activity of mitochondrial complexes I, II and IV. The effects of 50 µg/mL of the EAF on the growth of yeast cultures (growth curves in Figure 4) suggest that an exposure over a period of 8 h should promote toxic effects without significant cell death, which would be detectable by a decrease in values of optical density. The results obtained with mitochondria isolated from non-treated and treated yeast cultures are displayed in Figure 5A–D. To exclude the influence of putative differences in the degree of purity of the mitochondria obtained from different yeast cultures, the activity of mitochondrial complexes was normalized by the citrate synthase (CS) activity. As shown in Figure 5D, mitochondria obtained from the non-treated and treated yeast cultures exhibited similar CS activity, expressed as nmol of substrate/min/mg of protein. Since CS is often considered to be a rate limiting enzyme of the citric acid cycle, these data also suggest that the EAF does not impact the ability of yeast mitochondria to process pyruvate in the citric acid cycle. The data in Figure 5A–C show that yeast exposure to the EAF significantly decreases the activity of mitochondrial respiratory complexes I and II without detectable effects on the activities of complex IV.

To evaluate if the decreased activities of complexes I and II detected in mitochondria isolated from yeasts exposed to the *G. senegalensis* EAF can emerge from inhibition of these mitochondrial complexes, direct effects on their activity were assessed in mitochondria isolated from yeast control cultures. The obtained results (Figure 5E,F) show that the EAF decreases the activities of both complexes I and II in a concentration-dependent manner. However, the minimal concentration of the EAF required to promote a significant decrease in the complex I activity (25 µg/mL) is lower than that required to inhibit complex II (50 µg/mL). Although these in vitro results are qualitatively in line with those obtained by ex vivo assays, using mitochondria isolated from yeast cultures exposed to 50 µg/mL of the EAF (Figure 5A,B), their quantitative effects are divergent since the ex vivo assays show that complex II is more sensitive. Combining the results from Figure 5 with those from Figure 4, it can be suggested that the EAF includes compounds with the capability to cross several biological barriers to reach the mitochondria, thereby promoting specific inhibition at levels of complexes I and II. Considering the EAF chemical profile, it is expected that the lipophilic compounds that dominate this fraction, namely guieranone A and chlorophyll-related compounds, can be absorbed by yeasts and target the mitochondria via passive diffusion through the lipid membranes. It is well known that the mitochondrial redox chain is supplied by complex I NADH oxidation and by succinate processing at the level of complex II activity [33]. Therefore, with the inhibition of both complexes I and II, the EAF may reduce the coupling efficiency between the carbon flux (citrate cycle in mitochondrial matrix) and electron flux through the respiratory chain in the inner mitochondrial membrane, thereby worsening mitochondrial functionality and the yeasts’ ability to generate ATP by oxidative phosphorylation. Thus, the antifungal activity exhibited by the *G. senegalensis* EAF can emerge, total or partially, by mitochondrial intoxication.

### 2.6. G. senegalensis EAF Inhibits the Activity of Tyrosinase

Tyrosinase, a key enzyme in melanin biosynthesis, plays an important role in fungal pathways underlying spore formation, as well as in mechanism of defense against external factors, including UV radiation, toxic oxidant agents, dehydration and extreme temperatures [21]. Thus, this enzyme is considered a valuable target in fungicide development programs [34]. According to the Fungicide Resistance Action Committee (FRAC), the risk of drug resistance development in fungi pathogens targeted through single-site mechanism is significantly higher than for molecules or mixtures of molecules with multiple sites of action [35]. Since resistance development has been reported for many antifungal agents that work as mitochondrial toxins [36,37], the EAF ability to inhibit tyrosinase enzyme was assessed for its potential as a multi-target fungicide.

Figure 6A shows the activity of mushroom tyrosinase in the absence or presence of increasing concentrations of the EAF, using L-DOPA as the substrate at low (close of enzyme Km) and high concentrations. At a low substrate concentration (90 µM of L-DOPA), EAF decreased the activity of the enzyme in a concentration-dependent manner, with an IC_50_ of 48.86 ± 4.40 µg/mL. At a high substrate concentration (900 µM—a concentration that saturates the enzyme), the IC_50_ decreased to 18.04 ± 2.83 µg/mL. Thus, the *G. senegalensis* EAF is an effective inhibitor of mushroom tyrosinase. Although the IC_50_ value is a parameter largely used to screen enzyme inhibitors, it is highly dependent on the assay conditions, hindering the comparison among laboratories. Thus, the assessment of IC_50_ values at low and high enzyme substrate concentrations aims to improve the potential of this parameter to identify and compare enzyme inhibitors. In fact, previous works have reported that extracts of *G. senegalensis* leaves obtained with water, ethanol/water mixtures, methanol, or acetone have the ability to inhibit tyrosinase [38,39]. This bioactivity was attributed to phenolic compounds since the extract obtained with acetone exhibited strong tyrosinase inhibition and greater richness in phenolic compounds [39]. However, *G. senegalensis* acetone extract may have many other compounds in addition to phenolic compounds, weakening the reported relationship. Comparison of the present IC_50_ values with those previously reported is also difficult since different enzyme concentrations were used and the previous IC_50_ values were evaluated at a single substrate concentration.

To investigate the kinetic model of enzyme inhibition and confirm the effective ability of the EAF to inhibit the enzyme, tyrosinase activity supported by seven substrate concentrations was assessed in the absence (control) and presence of the EAF at 10 and 25 µg/mL. These concentrations were selected since they are in the range where the EAF exhibits mitochondrial toxicity and antifungal activity. As shown in Figure 6B,C, the enzyme activity as a function of substrate concentration was analysed by the Michaelis–Menten kinetic model. In fact, the Michaelis–Menten kinetic equation fits reasonably to the experimental data, allowing the determination of enzyme Vmax and Km, the apparent kinetic parameters (Figure 6D). Thus, a value of Km of 164.9 ± 29.25 µM was found for mushroom tyrosinase at 30 °C, which is in line with previously reported values [40]. Under our assay conditions, an enzyme Vmax value of 1.401 ± 0.008 nmol dopachrome/min was calculated. The *G. senegalensis* EAF at 10 or 25 µg/mL significantly decreased the enzyme Vmax (*p* ≤ 0.01) and increased enzyme Km, but for this kinetic parameter the differences only reached statistical significance for 25 µg/mL of the EAF. A decrease in Vmax and an increase in Km suggests that the tyrosinase inhibition promoted by the EAF can be described by a mixed model inhibition, with a predicted K_i_ value of 14.40 ± 4.49 µg/mL that is based on the equation generated by the GraphPad Prism software for this type of inhibition (Figure 6D). The K_i_ value is close to the IC_50_ value obtained with a substrate concentration that saturates the active center of the enzyme (i.e., 900 µM), suggesting predominance of non-competitive inhibition. Thus, the first tyrosinase inhibitory effect of the EAF can be associated with allosteric modulation of enzyme activity by compounds that do not compete with the substrate by the active site.

## 3. Materials and Methods

### 3.1. Standards and Reagents

Methanol, acetonitrile, and 2-propanol were obtained from Merck (Darmstadt, Germany). Formic acid was acquired from Carlo Erba Reagents S.A.S. (Val de Reuil, France). Dimethyl sulfoxide (DMSO) was purchased from Fischer Scientific (Loughborough, UK). Ethyl acetate was acquired from Valente e Ribeiro, Lda. (Belas, Portugal). Ethanol was acquired from VWR Chemicals (Radnor, PA, USA). *tert*-Butyl methyl ether was obtained from Honeywell (Charlotte, NC, USA). Gallic acid, myricetin-3-*O*-glucoside, myricetin-3-*O*-rhamnoside, quercetin-3-*O*-galactoside, quercetin-3-*O*-glucoside, quercetin-3-*O*-arabinoside, quercetin-3-*O*-rhamnoside, myricetin, kaempferol-3-*O*-glucoside, quercetin, hesperetin, isorhamnetin, penconazole, rotenone, mushroom tyrosinase, potassium cyanide, antimycin A, nicotinamide adenine dinucleotide (NADH), decylubiquinone, 2,6-dichlorophenolindophenol (DCPIP), oxaloacetate, succinate, cytochrome C, 5,5′-dithiobis(2-nitrobenzoic acid) (DTNB), acetyl-CoA, chlorophyll *a*, chlorophyll *b*, select yeast extract, and 3-morpholinepropanesulfonic acid (MOPS) were obtained from Sigma-Aldrich (St. Louis, MO, USA). Fluconazole was acquired from Alfa Aesar (Haverhill, MA, USA). RPMI-1640 broth medium (with L-glutamine, without bicarbonate and with the pH indicator phenol red) was purchased from Biochrom AG (Berlin, Germany). Sabouraud dextrose agar (SDA) and Sabouraud dextrose broth (SDB) were purchased from Bio-Mèrieux (Marcy L’Etoile, France).

### 3.2. Plant Material

*G. senegalensis* leaves were collected in Orango Island (Guinea-Bissau) in June 2017, February 2020, and November 2021. Voucher specimens were deposited at LAQV/REQUIMTE, University of Porto. Samples of leaves were air-dried, ground to a fine powder, and sieved to collect solid particles with a mean size less than 910 µm.

### 3.3. Preparation and Fractionation of G. senegalensis Leaves Crude Methanol Extract

For each collected *G. senegalensis* leaves sample, a CME was obtained through the maceration of 3 g of powdered plant material in 500 mL of methanol under stirring (200 rpm) for 2 h, followed by 30 min of sonication and an additional step of stirring for 1 h. The solid material was removed by centrifugation (1792× *g*, 4 min), and the liquid fraction was filtered through a Büchner funnel Witeg Germany^®^ to obtain the CME extract. The obtained methanol solution was subjected to reduced pressure in a Büchi Labortechnik AG Rotavapor^®^ R-215 (Flawil, Switzerland) at 40 °C to obtain the dried CME. The dry weight of the extract was determined, and the extraction yield was calculated. The samples were then preserved in vials protected from light for further studies. The solid-phase extraction (SPE) clean-up procedure using Chromabond C18 non-end-capped columns was applied to the CME in order to obtain two fractions, EAF and MF, and discard the water soluble compounds without affinity for the column, as previously described [41]. EAF and MF were dried by a nitrogen stream at room temperature, weighed to determine the fractionation yield, and preserved in vials protected from light for further studies. Three independent extractions were performed. 

### 3.4. General Chemical Characterization

The CME, EAF and MF were characterized in terms of total levels of: (i) Folin–Ciocalteu reactive phenolic compounds, using gallic acid as standard (range 0–14.2 µg/mL) [9]; (ii) sugars by the phenol-sulfuric acid procedure, using glucose as a standard (range 0–25 nmol) [42]; (iii) phospholipids by the Barlett and Lewis procedure after samples were subjected to acid hydrolysis at 180 °C, for 2 h, using phosphatidylcholine C16:0/18:1 as a standard (0–250 nmol) [12,43]; (iv) phytosterols by the Liebermann–Burchard procedure, using stigmasterol as a standard (0–250 µg) [44]; and (v) chlorophylls by UV-visible spectrum signatures at 665 nm for the CME, EAF and MF ethanol solutions using chlorophyll *a* as a standard (0–25 µg/mL) [14]. The presence of alkaloids in the CME, EAF and MF was assessed by the general alkaloid precipitation tests using Dragendorff’s (solution of potassium bismuth iodide), Mayer’s (potassium mercuric iodide solution) and Bertrand’s (silicotungstic acid solution) reagents [45].

### 3.5. Characterization of the Phenolic Profile

#### 3.5.1. HPLC-DAD-ESI(Ion Trap)-MS^n^ and UPLC-ESI-QTOF-MS^2^ Analyses

HPLC-DAD-ESI(Ion Trap)-MS^n^ analyses were carried out as described in Ferreres et al. [46], but with variations in the gradient; the mobile phase consisted of two solvents, water-formic acid (1%) (A) and acetonitrile (B), starting with 5% B and using a gradient to obtain 20% B at 10 min and 50% B at 25 min. The flow rate was 0.8 mL/min and the injection volume 20 µL. Chromatograms were recorded at 280, 320, 350, and 370 nm, and the mass full scan covered the range from *m*/*z* 100 to 1500.

Determination of the exact mass was carried out using an Agilent 1290 Infinity LC system coupled to the 6550 Accurate-Mass QTOF (Agilent Technologies, Waldbronn, Germany) with an electrospray interface (Jet Stream Technology). Samples (2 µL) were injected onto a reverse-phase Kinetex column (1.7 µm, C18, 100 Å, 50 × 2.1 mm; Phenomenex, Macclesfield, UK) with a SecurityGuard ULTRA Cartridge of the same material operating at 30 °C and a flow rate of 0.5 mL/min. The mobile phase was a mixture of acidified water (0.1% formic acid) (A) and acidified acetonitrile (0.1% formic acid) (B). Compounds were separated using the following gradient conditions: 0 min 5% B, obtaining 50% B at 20 min, and 70% B at 22 min. The optimal conditions for the electrospray interface were the following: gas temperature 280 °C, drying gas 11 L/min, nebulizer pressure 45 psi, sheath gas temperature 400 °C, and sheath gas flow 12 L/min. The MS system was operated in negative ion mode with the mass range set at *m*/*z* 100–1500 in full scan resolution mode. Further conditions were as described in [47].

#### 3.5.2. HPLC-DAD Analysis

Quantitative analysis was performed using the gradient and the mobile phases described for the HPLC-DAD-ESI(Ion Trap)-MS^n^ analyses. The dried extract obtained from the leaves of *G. senegalensis* was dissolved in methanol and filtered through a PTFE (0.45 µm pore size) membrane (Millipore, Bedford, MA, USA), 20 µL being injected. Triplicate analyses were performed on a Gilson HPLC-DAD unit (Gilson Medical Electronics, Villers le Bel, France) equipped with a reverse-phase Kinetex column (5 µm, C18, 100 Å, 150 × 4.6 mm; Phenomenex, Torrance, CA, USA). Detection was achieved with an Agilent 1260 series diode array detector (Agilent Technologies, Waldbronn, Germany). Spectral data from all peaks were collected in the range of 190–700 nm, and chromatograms were recorded at 280, 320, 350, and 370 nm. The data were processed on the Clarity software system, version 5.04.158 (DataApex, Ltd., Prague, Czech Republic).

Calibration curves were built with six concentrations of the external standards injected in triplicate. Myricetin-3-*O*-glucoside (**14**), myricetin-3-*O*-rhamnoside (**19**), quercetin-3-*O*-galactoside (**20**), quercetin-3-*O*-glucoside (**21**), quercetin-3-*O*-arabinoside (**23**), quercetin-3-*O*-rhamnoside (**24**), myricetin (**25**), quercetin (**28**), hesperetin (**32**) and isorhamnetin (**33**) were quantitated as themselves. Compounds **1**–**10** were quantitated against the standard curve of gallic acid; **11–13**, **15** and **18** were quantitated as myricetin-3-*O*-glucoside, **16** and **17** as quercetin-3-*O*-galactoside, **22** as quercetin-3-*O*-arabinoside, and **26**, **27**, **29**–**31** as kaempferol-3-*O*-glucoside. Linearity was assessed from the coefficients of determination (R^2^) of the calibration curves (Appendix A). Limits of detection (LOD) and quantification (LOQ) were determined from the residual standard deviation (σ) of the regression curves and the slopes (S), according to the equations LOD = 3.3σ/S and LOQ = 10σ/S (Appendix A). Further details can be found in the Appendix A.

### 3.6. Characterization of the Chlorophyll Profile

#### 3.6.1. HPLC-DAD Analysis

Dried samples of EAF were redissolved in ethyl acetate, filtered through a 0.45 μm pore size membrane and analysed on a Gilson HPLC-DAD unit (Gilson Medical Electronics, Villiers le Bel, France) equipped with a C30 YMC carotenoid column (5 μm, 250 × 4.6 mm; YMC, Kyoto, Japan). A gradient elution was conducted using a flow rate of 0.9 mL/min and an injection volume of 20 μL. The mobile phase consisted of two solvents: methanol (A), and tert-butyl methyl ether (B). It started with 95% A and used a gradient to obtain 70% at 30 min, 50% at 50 min, 0% at 60 min, 0% isocratic at 65 min and 95% at 70 min. Chromatograms were recorded at 420 and 450 nm and spectral data from all peaks were collected in the range of 190–700 nm. For quantification, calibration curves were built with six concentrations of the external standards injected in triplicate. Chlorophyll *a* and chlorophyll *b* were quantitated as themselves. Chlorophyll *a* derivatives were quantitated as chlorophyll *a*, and chlorophyll *b* derivatives were quantitated as chlorophyll *b*. Linearity was assessed from the coefficients of determination (R^2^) of the calibration curves (Appendix A). Limits of detection (LOD) and quantification (LOQ) were determined from the residual standard deviation (σ) of the regression curves and the slopes (S), according to the equations LOD = 3.3σ/S and LOQ = 10σ/S (Appendix A). Further details can be found in the Appendix A.

#### 3.6.2. UPLC-ESI-QTOF-MS^2^ Analyses

Analysis was performed in an Orbitrap Exploris 120 mass spectrometer (Thermo Fischer Scientific, Bremen, Germany) controlled by Orbitrap Exploris Tune Application 2.0.185.35 and Xcalibur 4.4.16.14 software. Samples (2 µL) were injected onto a reverse-phase C30 YMC carotenoid column (5 μm, 250 × 4.6 mm; YMC, Kyoto, Japan) with a SecurityGuard ULTRA Cartridge of the same material operating at 30 °C and a flow rate of 0.3 mL/min. The mobile phase consisted of two solvents: methanol (A) and *tert*-butyl methyl ether (B). It started with 100% A and used a gradient to obtain 30% at 90 min, 30% isocratic from 90 to 12min, and 100% A at 135 min. The capillary voltage of the electrospray ionization source (ESI) was set to 3.5 and 2.5 kV for positive and negative modes. The capillary temperature was 350 °C. The sheath, auxiliary and sweep gases flow rates were 50, 10 and 1 (arbitrary units, as provided by the software settings) with a 70% tube lens setting. The resolution of the MS scan was 60,000. Data-dependent MS/MS was performed on HCD using nitrogen as gas with collision energy settings of 35 V at 1500 of resolution. The range was set from 150 to 2000 Da. The MS system was operated in both positive and negative ion modes with the mass range set at *m*/*z* 100–1500 in full scan resolution mode.

### 3.7. Antifungal Activity Assessment

#### 3.7.1. Fungal Organisms

Four yeast strains (*C. albicans* (ATCC 10231), *C. krusei* (ATCC 6258), *C. neoformans* (CECT1078), *S. cerevisiae* (FF180)) and three filamentous fungi (*A. fumigatus* (ATCC 204305) *F. oxysporum* (FF/T-9) and *C. gloeosporioides* (FF/T-6)) were used in the present work. Yeast strains and filamentous fungi were stored in SDB with 20% glycerol at −80 °C. Before each test, a sub-culture in SDA of 1–5 days was prepared to achieve optimal growth conditions and purity.

#### 3.7.2. Assessment of Antifungal Activity by Susceptibility Tests and Time-Dependent Growth Effects

Minimum inhibitory concentrations (MICs) were determined by the broth microdilution method in accordance with the Clinical and Laboratory Standards Institute (CLSI) reference documents M27-A3, S3, and M38-A2, for yeasts and filamentous fungi, respectively.

Fungal organisms previously cultured in SDA were suspended in RPMI-1640 broth buffered with MOPS (pH 7.0). Two-fold serial dilutions of CME and of the respective fractions were prepared in RPMI-1640 broth. The solutions and cell suspensions were distributed into sterile 96-well plates that were incubated aerobically in a humid atmosphere, without agitation, at 35 °C and 25 °C, for yeasts and for filamentous fungi, respectively. The time of incubation was 48 h for *S. cerevisiae*, *C. albicans* and *C. krusei*, and 72 h for *C. neoformans* and for filamentous fungi. MIC was defined as the lowest concentration showing 90–100% absence of growth on the 96-well plates. Minimum fungicidal concentrations (MFCs) were determined by subculturing of 20 µL of all MIC plate wells without visible growth in SAB plates after 48 or 72 h of incubation, at 35 or 25 °C according to the fungi strain, as described above. The MFC was defined as the lowest concentration showing a total absence of growth on the SAB plates. For each assay condition, at least three independent experiments were performed. The medians of the obtained results in each type-test were used as the MIC and MFC values.

The effects of the CME, EAF and MF on the time-dependent growth of *S. cerevisiae* and *C. neoformans* liquid cultures were also assessed to evaluate the antifungal activity by changes in the specific growth rate. *S. cerevisisae* and *C. neoformans* cultured in SDA were subcultured in lactone-rich media and liquid SAB, respectively, and grew overnight, at 30 °C, under orbital shaking (200 rpm) to obtain an active inoculum. Then, test tubes with the suitable yeast strain culture medium without or the addition of the *G. senegalensis* CME, EAF or MF at 50 µg/mL were inoculated with the active inoculum and incubated for 60 h under the conditions mentioned above. The yeast growth was followed by changes in the optical density at 610 nm measured at different time points. For both yeast strains, the effects of increasing concentrations of *G. senegalensis* EAF (5–50 µg/mL) were also tested.

### 3.8. Assessment of the Effects on Mitochondria by Measuring the Mitochondrial Respiratory Chain Complex Activities

Yeast cultures of *S. cerevisiae* grown in 250 mL flasks with 50 mL of liquid medium in the absence (control) or presence of 50 µg/mL of EAF for 8 h were used to isolate mitochondria. *S. cerevisiae* were collected from the culture medium by centrifugation (1000× *g* for 4 min) and resuspended in 2 mL of buffer (250 mM sucrose, 5 mM HEPES buffer, pH = 7.4) at 4 °C. The cell wall of the yeast cells was mechanically disrupted by zirconia beads (1 g, 0.5 mm of diameter) under vortex stirring (five cycles, 1 min of stirring followed by a resting step for 2 min at 4 °C). The cell suspensions were filtered through a standard surgical gauze to remove the zirconia beads and the filtrate was homogenized at 4 °C in a Glass-Teflon Potter Elvejhem (clearance of 0.004–0.006 in.) using 20 up- and downstrokes at 500 rpm to promote cell lysis. Then, the cell lysates were centrifuged at 1000× *g* for 10 min at 4 °C, and the supernatant was collected. The supernatant was then centrifuged at 12,000× *g* for 15 min (4 °C), and the mitochondria-rich fraction was collected in the pellet. The pellet was resuspended in 250 µL of buffer medium (130 mM sucrose, 50 mM KCl, 5 mM MgCl_2_, 5 mM KH_2_PO_4_, 5 mM HEPES, pH = 7.4). An aliquot was used to determine the mitochondrial protein concentration by the biuret method, using bovine serum albumin as a standard reference [12], and the samples were then frozen in liquid nitrogen and stored at −80 °C for further studies.

The *S. cerevisiae* mitochondrial suspensions were submitted to three cycles of freezing/thawing before being used to assess the activity of mitochondrial complexes I, II and IV and citrate synthase at 30 °C using the standard procedures previously described [12]. The activity of the mitochondrial redox chain complexes was normalized by the activity of citrate synthase. Mitochondria obtained from yeasts cultured in control conditions were also used to evaluate the direct effects of increasing concentrations of the EAF on the activity of mitochondrial complexes I and II after 5 min of incubation. The results are expressed as a percentage of the control.

### 3.9. Tyrosinase Activity

The activity of mushroom tyrosinase was assessed by the oxidation of L-DOPA (substrate) by recording over 3 min the increase in absorbance at 475 nm resulting from dopachrome generation. Assays, in the absence or presence of increasing concentrations of *G. senegalensis* EAF, were carried out at 30 °C in 200 µL of 50 mM potassium phosphate monobasic buffer (pH 6.8) supplemented with 5 U/mL of enzyme and 90 or 900 µM of L-DOPA to assess the IC_50_, or with seven different substrate concentrations for enzyme kinetic studies. Enzyme activity was expressed in nmol/min using 3600 M^−1^ cm^−1^ as the extinction coefficient of dopachrome. Kojic acid was used as reference tyrosinase inhibitor.

### 3.10. Statistical Analysis

Statistical analysis was performed using GraphPad Prism 8 software. Data are presented as the mean ± standard deviation of at least three independent experiments or as medians. The level of significance between different groups was determined by one-way ANOVA with Bonferroni’s post-test. Values of *p* ≤ 0.05 were considered statistically significant.

## 4. Conclusions

From the overall results presented in this study, it can be concluded that *G. senegalensis* leaves have high potential as a renewable resource for the development of new multi-target fungicide formulations. The chemical characterization revealed that the CME of *G. senegalensis* leaves is a complex mixture of compounds containing sugars, phospholipids, phytosterols, guieranone A, porphyrin-containing compounds, and a great diversity of phenolic compounds, mainly flavonol 3-*O*-glycosylated derivatives. In fact, six compounds were identified in *G. senegalensis* leaves for the first time (two myricetin derivatives, two quercetin derivatives and, at least, two kaempferol derivatives). Although the *G. senegalensis* leaves CME exhibited weak antifungal activity against the three filamentous fungi strains tested, its EAF showed greater bioactivity against filamentous fungi. Moreover, this lipophilic fraction had excellent antifungal activity against pathogenic yeast strains, such as *C. neoformans* and *C. krusei*. In fact, the antifungal power of the EAF against these pathogenic yeasts was in the same order of magnitude as the antifungal activity of penconazole, which is used in agriculture, and fluconazole, which is used clinically. The chemical characterization revealed that guieranone A and the chlorophyll derivatives were retained in the EAF, confirming the association of the antifungal activity with these lipophilic compounds and excluding the association with phenolic compounds. The mechanisms of action underlying the antifungal activity were also revealed using *S. cerevisiae*, a suitable yeast model, since the EAF exhibits similar bioactivity against *S. cerevisiae* and *C. neoformans*. It was found that the EAF has the ability to inhibit the yeast mitochondrial respiratory complexes I and II and the tyrosinase enzyme in a concentration range that exhibits in vivo effects. Thus, the antifungal activity of the EAF against yeasts emerges from specific mitochondrial toxicity together with disruption of the biochemical pathways dependent on tyrosinase activity. Therefore, the present work highlights the potential of *G. senegalensis* leaves for the development of innovative multi-target antifungal formulations.

## Figures and Tables

**Figure 1 antibiotics-12-00869-f001:**
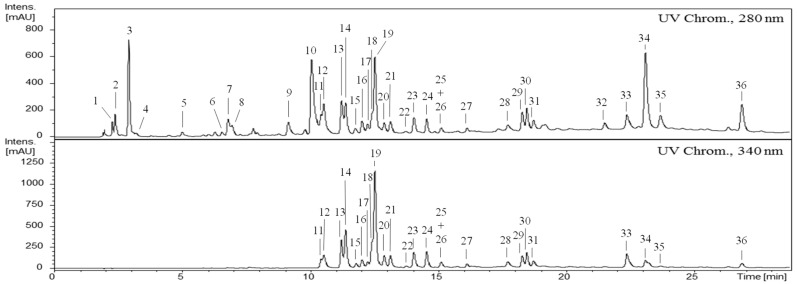
HPLC-UV profile of the crude methanol extract (CME) obtained from leaves of *G. senegalensis*. Polyphenolic compounds were analysed by reverse-phase HPLC-DAD-ESI-MS^n^ using a C18 column with spectra acquisition in continuous scan mode from 200 to 600 nm. The chromatograms were recorded at 280 and 340 nm. The compound identifications are indicated in Table 2 and in the text.

**Figure 2 antibiotics-12-00869-f002:**
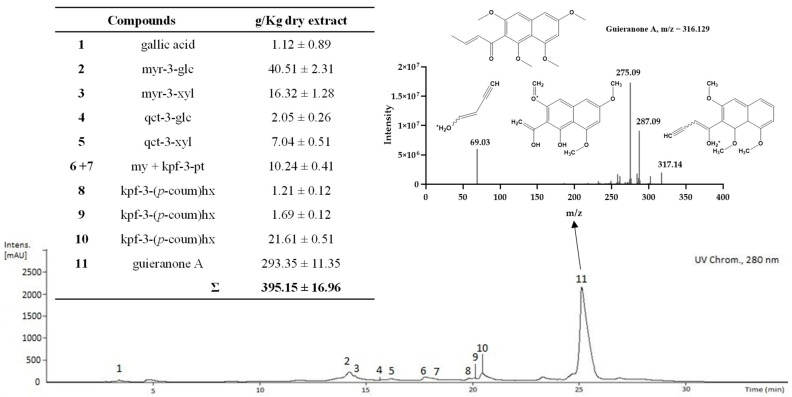
HPLC-UV profile of the ethyl acetate fraction (EAF) obtained by solid-phase extraction (HyperSep C18) from the *G. senegalensis* leaves CME. Polyphenolic compounds were analysed by reverse-phase HPLC-DAD using a C18 column with spectra acquisition in continuous scan mode from 200 to 600 nm. The chromatogram was recorded at 280 nm.

**Figure 3 antibiotics-12-00869-f003:**
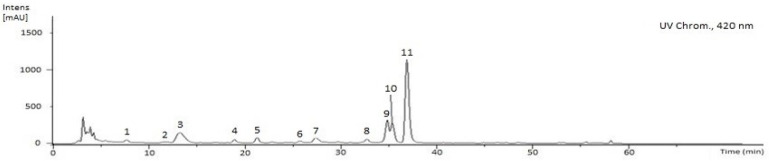
HPLC-UV profile of the EAF obtained by solid-phase extraction (HyperSep C18) from the *G. senegalensis* leaves CME, using a C30 column with spectra acquisition in continuous scan mode from 200 to 600 nm and the chromatogram recorded at 420 nm. Chlorophylls were also analysed by reverse-phase HPLC-ESI-MS^n^. The compound identification is indicated in Table 3 and the text.

**Figure 4 antibiotics-12-00869-f004:**
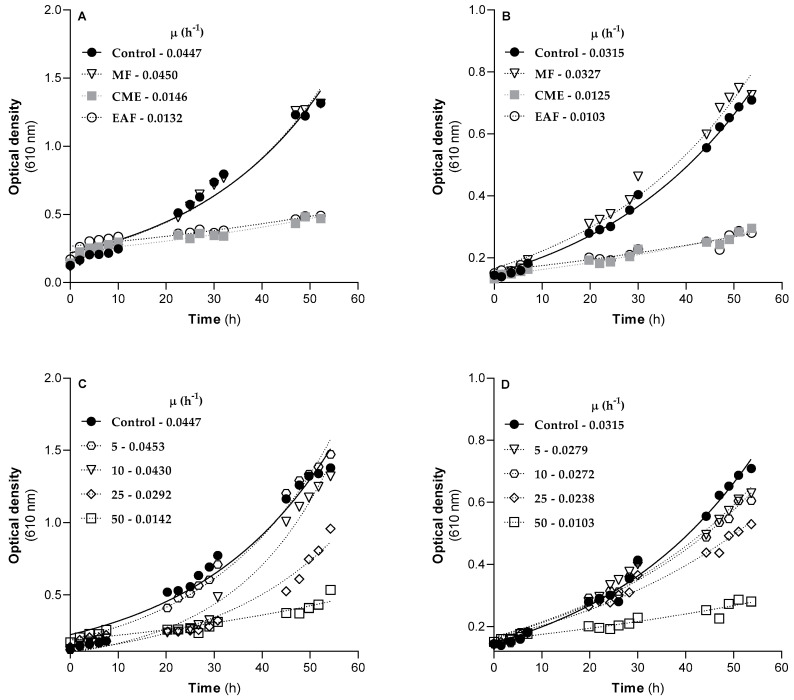
Effects of *G. senegalensis* leaves CME and of its MF and EAF on the growth of *S. cerevisiae* (**A**,**C**), and *C. neoformans* (**B**,**D**), at 30 °C, in the respective standard media. Time-dependent growth of *S. cerevisiae* (**A**) and *C. neoformans* (**B**) cultures in the absence and presence of 50 µg/mL of *G. senegalensis* leaves CME or in the presence of its fractions, MF and EAF, at 50 µg/mL. Effects of increasing concentrations of EAF on *S. cerevisiae* (**C**) and *C. neoformans* (**D**) growth. The cell number was measured by the optical density at 610 nm. The results shown are representative of three independent experiments using three started yeast cultures.

**Figure 5 antibiotics-12-00869-f005:**
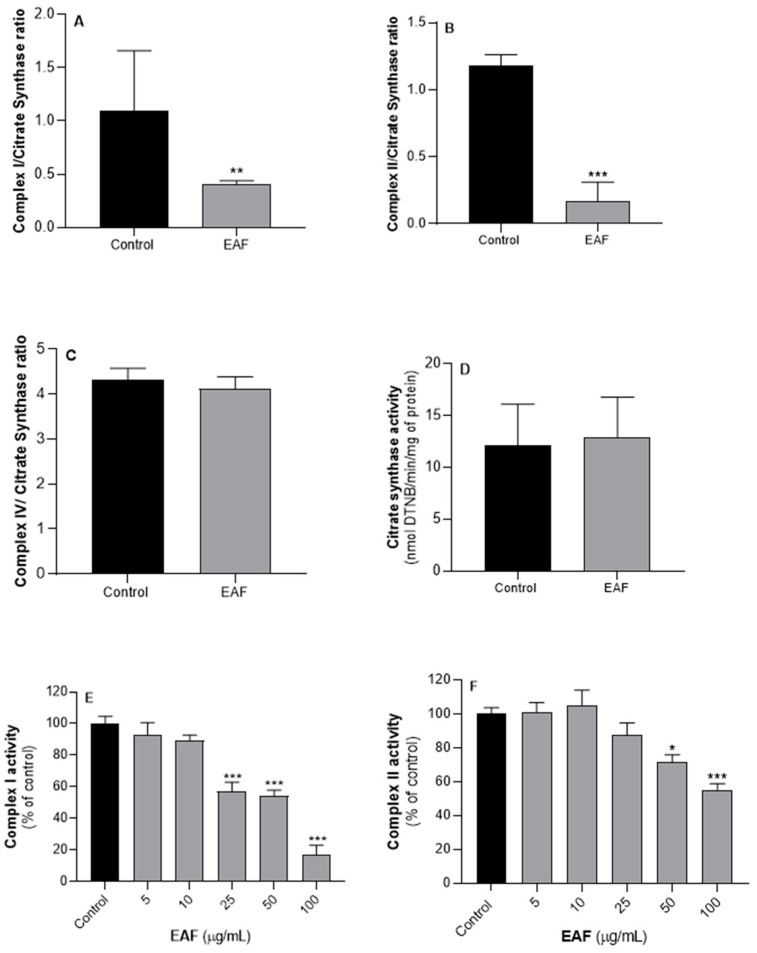
Effects of the exposure of *S. cerevisiae* cultures to 50 µg/mL of *G. senegalensis* EAF, for 8 h, on the activity of the mitochondrial redox complexes I, II, and IV and citrate synthase (**A**–**D**). The activity of mitochondrial redox chain complexes was normalized by citrate synthase activity. In vitro effects of *G. senegalensis* EAF on the activity of complexes I and II of mitochondria isolated from *S. cerevisiae* cultures grown in control condition, i.e., in the absence of EAF (**E**,**F**). Results are presented as mean ± SD of three independent assays. *, **, *** Significantly different from control condition, with *p* ≤ 0.05, *p* ≤ 0.01 and *p* ≤ 0.001, respectively. Black bars—control; gray bars—EAF treatment.

**Figure 6 antibiotics-12-00869-f006:**
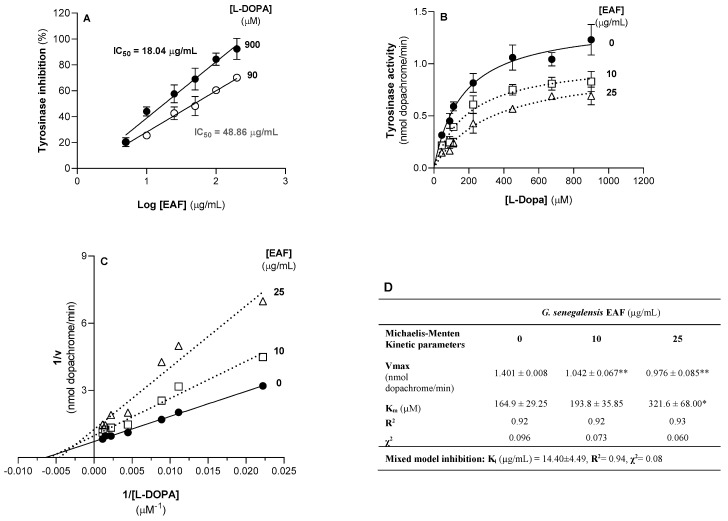
Effects of increasing concentrations of *G. senegalensis* EAF on the activity of pure mushroom tyrosinase assessed in the presence of low (90 µM) and high (900 µM) L-Dopa (enzyme substrate) concentration (**A**). Kinetics of tyrosinase activity in the absence and presence of 10 or 25 µg/mL of *G. senegalensis* EAF using seven concentrations of substrate (**B**). Data were fitted into a Michaelis-Menten kinetic equation and used to construct the double-reciprocal plot analysis (**C**). Table with the apparent Michaelis–Menten kinetic parameters and Ki values obtained with a non-competitive inhibition model using GraphPad Prism software (**D**). *, ** Significantly different from control condition, with *p* ≤ 0.05, *p* ≤ 0.01, respectively.

**Table 1 antibiotics-12-00869-t001:** Extraction yields of *Guiera senegalensis* leaves with methanol (crude methanol extract (CME)) and of its solid-phase fractionation obtained with ethyl acetate (ethyl acetate fraction (EAF)) and methanol (methanol fraction (MF)). There is also a general characterization in terms of total levels of Folin–Ciocalteu reactive phenolic compounds, sugars, phospholipids, phytosterols, chlorophylls and alkaloids with the enrichment factor (E.F.) indicated for each compound.

(µg Compound Equivalents per mg Extract)	CME	EAF	E.F.	MF	E.F.
Extraction/fraction yield (%)	17.58 ± 5.43	24.15 ± 16.43	n. a.	11.44 ± 6.89	n. a.
Folin–Ciocalteu reactive phenolic compounds	37.63 ± 3.48	46.12 ± 5.98	1.23	17.7 ± 1.13	0.47
Total Sugars	58.91 ± 4.56	76.45 ± 20.90	1.30	33.70 ± 3.44	0.57
Phospholipids	37.70 ± 6.02	22.38 ± 6.20	0.59	8.45 ± 6.38	0.22
Phytosterols	67.39 ± 6.30	193.65 ± 20.38	2.87	17.17 ± 8.16	0.25
Chlorophylls	5.83 ± 1.00	28.63 ± 1.36	4.91	Not detected	n. a.
Alkaloids	Not detected	Not detected	n. a.	Not detected	n. a.

n. a.—not applicable.

**Table 2 antibiotics-12-00869-t002:** Identification and quantification (dry weight basis) of each phenolic compound detected in the *G. senegalensis* leaves CME by HPLC-DAD-ESI-MS^n^: Rt (retention time), UV (ultraviolet-visible absorption features), Molecular Formula, MS features in negative or positive (for guieranone) ion modes with indication of main MS^2^ detected fragments.

Compounds ^1^		Rt (min)	UV (nm)	Molecular Formula	[M−H]^−^, *m*/*z*	MS^2^ Fragments^−^, *m*/*z* (%)				g/Kg Dry Extract ^3^
			galloyl derivatives							
1	gall-quinc ac (GQA)	2.2	272	C_14_H_15_O_10_	343.0673	191.0565(Quinc-H), 169.0145(GA-H)				1.33 ± 0.01
2	digall-Quinc ac (dGQA)	2.4	272	C_21_H_19_O_14_	495.0780	343.0669(GQA-H), 191.0568(QA-H), 169.0149(GA-H)				2.06 ± 0.02
3	gallic ac (GA)	2.9	273	C_7_H_6_O_5_	169.0147	125.0244(GA-H-44)				11.64 ± 0.25
4	gall-quinc ac (GQA)	3.1	274	C_14_H_15_O_10_	343.0675	191.0563(Quinc-H), 169.0146(GA-H)				0.37 ± 0.004
5	digall-quinc ac (dGQA)	5.0	273	C_21_H_19_O_14_	495.0782	343.0670(GQA-H), 191.0569(QA-H), 169.0152(GA-H)				0.46 ± 0.02
6	digall-quinc ac (dGQA)	6.6	275	C_21_H_19_O_14_	495.0779	343.0668(GQA-H), 191.0569(QA-H), 169.0150(GA-H)				0.59 ± 0.03
7	trigall-quinc ac (tGQA)	6.8	272	C_28_H_23_O_18_	647.0892	495.0774(dGQA-H), 343.0670(GQA-H), 169.0147(GA-H)				2.25 ± 0.05
8	digall-quinc ac (dGQA)	6.9	275	C_21_H_19_O_14_	495.0780	343.0671(GQA-H), 191.0570(QA-H), 169.0146(GA-H)				1.60 ± 0.04
9	trigall-quinc ac (tGQA)	9.1	275	C_28_H_23_O_18_	647.0895	495.0770(dGQA-H), 343.0672(GQA-H), 307.0270, 169.0147(GA-H)				1.98 ± 0.006
10	tetragall-quinc ac	10.1	276	C_35_H_27_O_22_	799.1000	647.0894(tGQA-H), 629.0768(tGQA-H-W), 601.0818(tGQA-H-W-28), 495.0780(dGQA-H), 343.0665(GQA-H), 169.0145(GA-H)				11.48 ± 0.07
						Σ				33.76 ± 0.50
			flavonoid glycosides and aglycones							
						−152		[M−H/2H]^−^		
11	my-3-(gall)galc	10.4	264, 300sh, 356	C_28_H_24_O_17_	631.0917	479.0843(100)		316.0228(15)		5.36 ± 0.16
12	my-3-(gall)glc	10.5	264, 300sh, 355	C_28_H_24_O_17_	631.0911	479.0845(100)		316.0226(15)		1.45 ± 0.60
13	my-3-galc	11.2	254sh, 265, 300sh, 356	C_21_H_20_O_13_	479.8361			316.0229(100)		28.63 ± 0.53
14	my-3-glc	11.4	253, 264, 302, 356	C_21_H_20_O_13_	479.0834			316.0230(100)		29.64 ± 1.39
15	my-3-xyl	11.8	255sh, 265, 302sh, 354	C_20_H_18_O_12_	449.0716			316.0225(100)		5.09 ± 0.21
16	qct-3-(gall)galc	12.0	254sh, 266, 300sh, 356	C_28_H_24_O_16_	615.1025	463.0877(100)		301.0350(20)		2.63 ± 0.06
17	qct-3-(gall)glc	12.2	254sh, 265, 300sh, 355	C_28_H_24_O_16_	615.1018	463.0881(100)		301.0352(30)		1.86 ± 0.02
18	my-3-arab	12.4	254sh, 264, 302sh, 356	C_20_H_18_O_12_	449.0720			316.0227(100)		12.35 ± 0.07
19	my-3-rh	12.5	254, 264sh, 300sh, 352	C_21_H_20_O_12_	463.0882			316.0228(100)		36.94 ± 1.21
20	qct-3-galc	12.9	256, 266sh, 300sh, 354	C_21_H_20_O_12_	463.0876			301.0349(100)		7.00 ± 0.11
21	qct-3-glc	13.1	255, 266sh, 302sh, 354	C_21_H_20_O_12_	463.0880			301.0353(100)		21.06 ± 0.16
22	qct-3-xyl	13.7	255, 266sh, 302sh, 352	C_20_H_18_O_11_	433.0782			301.0350(100)		1.24 ± 0.04
23	qct-3-arab	14.0	255, 266sh, 300sh, 352	C_20_H_18_O_11_	433.0780			301.0351(100)		10.70 ± 0.25
24	qct-3-rh	14.5	255, 265sh, 302sh, 352	C_21_H_20_O_11_	447.0938			301.0348(100)		8.66 ± 0.07
25	my	15.2	---- ^2^	C_15_H_10_O_8_	317.0307					11.40 ± 0.04
26	kpf-3-pt	15.2	---- ^2^	C_20_H_18_O_10_	417.0831			284.0338(100)		
27	kpf-3-rh	16.1	266, 288sh, 348	C_21_H_20_O_10_	431.0975			285.0397(100)		1.87 ± 0.09
28	qct	17.7	256, 265sh, 300sh, 368	C_15_H_10_O_7_	301.0357					0.37 ± 0.02
32	hespt	21.5	285, 232sh	C_16_H_14_O_6_	301.0728					0.07 ± 0.002
33	isorhmn	22.4	256, 266sh, 298sh, 370	C_16_H_12_O_7_	315.0536					1.05 ± 0.03
								Σ		187.37 ± 5.06
			flavonols glycosides-cinnamoyl derivatives							
						−146				
29	kpf-3-(*p*-coum)hx	18.3	266, 296sh, 314, 350sh	C_30_H_25_O_13_	593.1301	447.0935(15)		284.0338(100)		2.16 ± 0.10
30	kpf-3-(*p*-coum)hx	18.5	265, 295sh, 314, 352sh	C_30_H_25_O_13_	593.1302	447.0935(10)		284.0340(100)		5.35 ± 0.23
31	kpf-3-(*p*-coum)hx	18.7	266, 294sh, 316, 350sh	C_30_H_25_O_13_	593.1299	447.0935(25)		284.0339(100)		1.11 ± 0.04
								Σ		8.62 ± 0.37
				other compounds—positive ion mode		[M+H]^+^, *m*/*z*				
34	guieranone A	23.1	266sh, 274, 312, 326, 380	C_18_H_20_O_5_	317.1290	287.0913(60)	275.0913(100)		69.0335(40)	36.46 ± 2.87
Σ										266.21 ± 8.79

^1^ my: myricetin; qct: quercetin; hespt: hesperetin; kpf: kaempferol; isorhmn: isorhamnetin; galc: galactoside; glc: glucoside; pt: pentoside; xyl: xyloside; arab: arabinoside; rh: rhamnoside; gall: gallic acid. ^2^ UV spectra could not be properly observed due to the co-elution of **25** and **26**. ^3^ Results correspond to mean ± standard deviation (n = 3).

**Table 3 antibiotics-12-00869-t003:** Identification and quantification (dry weight basis) of each chlorophyll detected in the *G. senegalensis* leaves EAF by HPLC-ESI-MS^n^: Rt (retention time), UV (ultraviolet-visible absorption features), Molecular Formula, MS features in positive ion mode with indication of main MS^2^ detected fragments.

	Compounds	Rt (min)	UV (nm)	Molecular Formula	[M+H]^+^ (*m*/*z*)	MS^2^[M+H]^+^, *m*/*z* (%)	g/Kg Dry Extract ^1^
1	HO-lactone-chlorophyll *a*	7.71	407, 664	C_55_H_72_MgN_4_O_7_	925.00	647.2269(60) 588.2128(20) 553.1991(35)	1.78 ± 0.34
2	chlorophyll *b*	11.91	435	C_55_H_70_MgN_4_O_6_	907.55	553.1992(30) 227.1603(100)	0.67 ± 0.03
3	HO-chlorophyll *a*	13.44	407, 665	C_55_H_72_MgN_4_O_6_	909.53	555.2154(100) 227.1608(20)	13.08 ± 1.63
4	chlorophyll *a*	18.84	418, 442, 470	C_55_H_72_MgN_4_O_5_	893.70	539.2418(100) 227.1598(30)	1.07 ± 0.18
5	pheophorbide *a*	21.19	418, 442, 471	C_35_H_36_N_4_O_5_	593.27	533.2544(80)	1.52 ± 0.40
6	HO-lactone-pheophytin *b*	25.68	434, 410	C_55_H_72_N_4_O_8_	917.70	830.5706(5) 639.2801(2) 579.2597(6) 551.2652(100)	0.63 ± 0.03
7	HO-lactone-pheophytin *a*	27.27	408, 664	C_55_H_74_N_4_O_7_	903.65	625.2648(20) 607.2546(20) 565.2444 (80)	2.50 ± 1.77
8	HO-pheophytin *b*	32.60	435, 412, 654	C_55_H_72_N_4_O_7_	901.65	623.2682(5) 563.2349 (5) 227.1575(100)	5.28 ± 2.89
9	pheophytin *b*	34.88	434, 412, 655	C_55_H_72_N_4_O_6_	885.55	826.5379(100) 547.2350(30) 535.2348(80)	12.67 ± 1.82
10	HO-pheophytin *a*	35.38	407, 666	C_55_H_74_N_4_O_6_	887.65	609.2703(40) 591.2598(60) 531.2389(100)	11.23 ± 0.72
11	pheophytin *a*	36.95	407, 665	C_55_H_74_N_4_O_5_	871.57	593.2754(100) 533.2546(80) 519.2385(5)	44.76 ± 1.78
						Σ	95.19 ± 11.59

^1^ Results correspond to mean ± standard deviation (n = 3).

**Table 4 antibiotics-12-00869-t004:** Antifungal activity of the *G. senegalensis* CME and its fractions, EAF and MF, against three species of filamentous fungi, assessed by MIC and MFC.

Filamentous Fungi	MIC ^1^ (µg/mL)			MFC (µg/mL)
	CME	EAF	MF	EAF
*Aspergillus fumigatus*	>2000	2000	>2000	>2000
*Fusarium oxysporum*	>2000	2000	>2000	>2000
*Colletotrichum gloeosporioides*	>2000	500	>2000	1000

^1^ Concentration that inhibited at least 90% of growth; CME—crude methanol extract; EAF—ethyl acetate fraction; MF—methanol fraction; MIC—minimal inhibitory concentration; MFC—minimal fungicidal concentration.

**Table 5 antibiotics-12-00869-t005:** Antifungal activity of the *G. senegalensis* EAF against four yeast strains, assessed by MIC and MFC. The fungicides penconazole and fluconazole were used as positive controls.

Yeast	MIC (µg/mL)			MFC (µg/mL)		
	EAF	Penconazole	Fluconazole	EAF	Penconazole	Fluconazole
*Sacharomyces cerevisiae*	8	1	8	500	1	16
*Candida albicans*	250	32	2	>1000	>32	>128
*Candida krusei*	16	16	32	500	≥32	>128
*Cryptococcus neoformans*	8	16	8	≥125	>16	>32

## Data Availability

The data presented in this study are available on request from the corresponding author.

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
