# Peer review of "Antifungal Activity of *Guiera senegalensis*: From the Chemical Composition to the Mitochondrial Toxic Effects and Tyrosinase Inhibition"

_antibiotics, 2023, doi:10.3390/antibiotics12050869_

Round 1
Reviewer 1 Report
Overall this work represents a comprehensive study on G senegalensis extract. The results presented in Table 1 have no implications in the finding of this work except the first line. This may be incorporated in text deleting Table one.
The manuscrript is acceptable.
overall language and writing abilities are good.
Author Response
Comment: “Overall this work represents a comprehensive study on G senegalensis extract. The results presented in Table 1 have no implications in the finding of this work except the first line. This may be incorporated in text deleting Table one. The manuscript is acceptable. Overall language and writing abilities are good.”
Answer to comment: First, we would like to acknowledge the manuscript appreciation made by the reviewer, and to recognize the scientific value of the recommendation. While we acknowledge the pertinence of the suggestion, we prefer to keep Table 1 as it gives an overview of the chemical composition not only of the CME but also on the fractions obtained by SPE fractionation, allowing to define the next steps of the work. In our opinion, Table 1 information has relevance for the interpretation of the obtained antifungal activity. For example, it allows ruling out alkaloids from the antifungal activity well evidenced by EAF. Additionally, the future EAF valorisation as an antifungal agent requires quality control parameters, and Table 1 information may constitute a useful tool in this matter. Despite western pharmaceutical and agrochemical industries prefer to develop drug formulations with single molecules, the use of extracts has high relevance for many African countries, including those where G. senegalensis is an abundant species.

Reviewer 2 Report
Dear Authors
After carefully reading your manuscript, I can say that it is very well structured and extensive. The title, discussions, and conclusion are detailed and logical, English is also good, with almost no mistakes.
I still have some minor comments on the manuscript:
1. At the end of the introduction, please emphasize more on the aim of the work, showing the novelty of the work.
2. Please add references to the following line numbers: 129 and 705.
3. In Table 1, E.F. for chlorophylls is 4.91, while in the text (line 132) is 4.92.
I think that your extensive work is worth to publish.
Regards
Author Response
Reviewer 2
General comment: “After carefully reading your manuscript, I can say that it is very well structured and extensive. The title, discussions, and conclusion are detailed and logical, English is also good, with almost no mistakes.”
Comment 1: “At the end of the introduction, please emphasize more on the aim of the work, showing the novelty of the work.”
Answer to comment 1: We would like to thank the reviewer for his/her comments, which allow to improve our paper. As suggested, the aim of the work was clarified, highlighting the novelty of the research, in final of the introduction section (lines 68-77), as follow:
“Working within this scientific scenario, a crude methanol extract (CME) of G. senegalensis leaves was prepared and fractionated on solid phase C18 column to obtain an ethyl acetate fraction (EAF) and a methanol fraction (MF). The chemical composition of CME, EAF and MF will be disclosed and their relationship with antifungal activity against several filamentous fungi and yeast strains will be discussed in the context of pest control and human health. Therefore, the present work aims to characterize if and how G. senegalensis leaves have antifungal activity, unveiling the mechanisms of action and stablishing a clear relationship between activity and chemical composition. It is the key to establishing the scientific bases that support the use of G. senegalensis leaves as a renewable source for the development of new fungicides.”
Comment 2: “Please add references to the following line numbers: 129 and 705.”
Answer to comment 2: As reviewer’s suggestion, a reference for chlorophyll quantification by UV-visible spectroscopy was included. Please see the new reference [14].
Comment 3: “In Table 1, E.F. for chlorophylls is 4.91, while in the text (line 132) is 4.92.”
Answer to comment 3: We apologize for the mistake, and the text was modified according with data in Table 1, as can be seen in page 4, line 137.

Reviewer 3 Report
Guiera senegalensis is a medicinal shrub widely distributed in African countries, which is commonly used for the treatment of several chronic diseases and infections. The authors declaimed that chemical contents of G. senegalensis leaves CME, MF and EAF have the ability to resist fungi. Through in vivo and in vitro experiments, the authors suggest that the antifungal activity of EAF against yeasts emerges from the specific mitochondrial toxicity together with the disruption of the biochemical pathways dependent of the tyrosinase activity, and provides a potential candidate of G. senegalensis leaves for the development of antifungal formulations.
Comments:
1. For further study, if the antifungal activity of EAF mainly comes from guieranone 891 A and the chlorophyll derivatives, then the antifungal effect of each one alone is encouraged to determined seperately to check which one dominates the effects. Does any other compenents may also have antifungal effects, besides these two kinds of components.
2. Experiments were conducted using plant leaves in this manuscript. Do extracts of other parts of Guiera senegalensis also have antifungal activity?
Guiera senegalensis is a medicinal shrub widely distributed in African countries, which is commonly used for the treatment of several chronic diseases and infections. The authors declaimed that chemical contents of G. senegalensis leaves CME, MF and EAF have the ability to resist fungi. Through in vivo and in vitro experiments, the authors suggest that the antifungal activity of EAF against yeasts emerges from the specific mitochondrial toxicity together with the disruption of the biochemical pathways dependent of the tyrosinase activity, and provides a potential candidate of G. senegalensis leaves for the development of antifungal formulations.
Comments:
1. For further study, if the antifungal activity of EAF mainly comes from guieranone 891 A and the chlorophyll derivatives, then the antifungal effect of each one alone is encouraged to determined seperately to check which one dominates the effects. Does any other compenents may also have antifungal effects, besides these two kinds of components.
2. Experiments were conducted using plant leaves in this manuscript. Do extracts of other parts of Guiera senegalensis also have antifungal activity?
Author Response
Reviewer 3
General comment: “Guiera senegalensis is a medicinal shrub widely distributed in African countries, which is commonly used for the treatment of several chronic diseases and infections. The authors declaimed that chemical contents of G. senegalensis leaves CME, MF and EAF have the ability to resist fungi. Through in vivo and in vitro experiments, the authors suggest that the antifungal activity of EAF against yeasts emerges from the specific mitochondrial toxicity together with the disruption of the biochemical pathways dependent of the tyrosinase activity and provides a potential candidate of G. senegalensis leaves for the development of antifungal formulations.”
Comment 1: “For further study, if the antifungal activity of EAF mainly comes from guieranone A and the chlorophyll derivatives, then the antifungal effect of each one alone is encouraged to determined separately to check which one dominates the effects. Does any other components may also have antifungal effects, besides these two kinds of components.”
Answer to comment 1: We recognize the scientific relevance of this comment. In fact, we are developing a new investigation to address this issue. For this purpose, the components of the EAF are being separated by preparative HPLC to separate not only guieranone A from the chlorophylls but also the different derivatives of the chlorophylls. In the next step, the activity of the components will be evaluated individually and in mixtures in order to identify the compounds that are responsible for the antifungal activity, and to characterize possible interactions (e.g., additive effects, synergism and potentiation). However, this research will be used to prepare a new article in near future.
Despite this, the relevance of the reviewer’s comment was included in the new version of the manuscript (page 15, lines 482-487), as follow:
“Therefore, the identification of the contribution of guieranone A and chlorophyll-related compounds for antifungal activity of EAF is an issue with scientific and pharmacological relevance that remains open. In fact, the addictive and synergistic effects as well potentiation and antagonism can play a key role on the activity revealed by a mixture of compounds, including plant extracts [31]. Thus, the minor components of EAF may also contribute for the detected antifungal activity.”
Comment 3: “Experiments were conducted using plant leaves in this manuscript. Do extracts of other parts of Guiera senegalensis also have antifungal activity?”
Answer to comment 3: We agree with the reviewer comment. In fact, in the manuscript are indicated the studies that report antifungal activity for extracts obtained from G. senegalensis galls and roots. Please see lines 57-59, page 2 and the new reference [6].

Reviewer 4 Report
The research for new compounds with antifungal and antimicrobial action is due to the development of resistance to conventional treatments which determines serious repercussions ranging from health consequences, crop yields, economic repercussions and represents a challenge for many researchers from multiple disciplines. The paper submitted by the Authors fits into this field of research, as they evaluate in vitro the activity of an extract (and subsequent fractionation) of an extract of Guiera senegalensis J. F. Gmel. a medicinal shrub widely distributed in African countries. an interesting study as the possibility of using compounds of natural origin to fight fungal and bacterial infections compared to synthetic compounds could also lead to fewer side effects and also favour a circular economy
The authors introduce their work in an exhaustive manner. The chemical part of the paper is extremely detailed, but still understandable even by the less expert. Cellular analyses on fungal species and biochemical investigations about mitochondrial functionality alteration and inhibition of tyrosinase are well designed and conducted, the explanation of the results is adequately supported and compared with the scientific literature. The materials and methods section is appropriate..
The conclusions are worthy of consideration and perspective also with subsequent insights into other fungal and even bacterial species. In my opinion the paper deserves the publication. Good paper.
Only minor editing of English language
Author Response
Reviewer 4
General Comment: “The research for new compounds with antifungal and antimicrobial action is due to the development of resistance to conventional treatments which determines serious repercussions ranging from health consequences, crop yields, economic repercussions and represents a challenge for many researchers from multiple disciplines. The paper submitted by the Authors fits into this field of research, as they evaluate in vitro the activity of an extract (and subsequent fractionation) of an extract of Guiera senegalensis J. F. Gmel. a medicinal shrub widely distributed in African countries. An interesting study as the possibility of using compounds of natural origin to fight fungal and bacterial infections compared to synthetic compounds could also lead to fewer side effects and also favour a circular economy. The authors introduce their work in an exhaustive manner. The chemical part of the paper is extremely detailed, but still understandable even by the less expert. Cellular analyses on fungal species and biochemical investigations about mitochondrial functionality alteration and inhibition of tyrosinase are well designed and conducted, the explanation of the results is adequately supported and compared with the scientific literature. The materials and methods section is appropriate. The conclusions are worthy of consideration and perspective also with subsequent insights into other fungal and even bacterial species. In my opinion the paper deserves the publication. Good paper.”
Answer to comment 1: Thank you for reviewing the manuscript, highlighting its positive aspects.
